# iPRESTO: Automated discovery of biosynthetic sub-clusters linked to specific natural product substructures

**Joris J. R. Louwen**[1], **Satria A. Kautsar**[1], **Sven van der Burg**[2], **Marnix H. Medema**[1]*, **Justin J. J. van der Hooft**[1,3]*

**1** Bioinformatics Group, Wageningen University, Wageningen, the Netherlands, **2** Netherlands eScience Center, Amsterdam, the Netherlands, **3** Department of Biochemistry, University of Johannesburg, Johannesburg, South Africa

* marnix.medema@wur.nl (MHM); justin.vanderhooft@wur.nl (JJJvdH)

**Data Availability Statement:** All relevant data are within the paper, its Supporting information files, and on Zenodo at https://doi.org/10.5281/zenodo.

## Abstract

Microbial specialised metabolism is full of valuable natural products that are applied clinically, agriculturally, and industrially. The genes that encode their biosynthesis are often physically clustered on the genome in biosynthetic gene clusters (BGCs). Many BGCs consist of multiple groups of co-evolving genes called sub-clusters that are responsible for the biosynthesis of a specific chemical moiety in a natural product. Sub-clusters therefore provide an important link between the structures of a natural product and its BGC, which can be leveraged for predicting natural product structures from sequence, as well as for linking chemical structures and metabolomics-derived mass features to BGCs. While some initial computational methodologies have been devised for sub-cluster detection, current approaches are not scalable, have only been run on small and outdated datasets, or produce an impractically large number of possible sub-clusters to mine through. Here, we constructed a scalable method for unsupervised sub-cluster detection, called iPRESTO, based on topic modelling and statistical analysis of co-occurrence patterns of enzyme-coding protein families. iPRESTO was used to mine sub-clusters across 150,000 prokaryotic BGCs from antiSMASH-DB. After annotating a fraction of the resulting sub-cluster families, we could predict a substructure for 16% of the antiSMASH-DB BGCs. Additionally, our method was able to confirm 83% of the experimentally characterised sub-clusters in MIBiG reference BGCs. Based on iPRESTO-detected sub-clusters, we could correctly identify the BGCs for xenorhabdin and salbostatin biosynthesis (which had not yet been annotated in BGC databases), as well as propose a candidate BGC for akashin biosynthesis. Additionally, we show for a collection of 145 actinobacteria how substructures can aid in linking BGCs to molecules by correlating iPRESTO-detected sub-clusters to MS/MS-derived Mass2Motifs substructure patterns. This work paves the way for deeper functional and structural annotation of microbial BGCs by improved linking of orphan molecules to their cognate gene clusters, thus facilitating accelerated natural product discovery.

6953657. All code is available at https://git.
wageningenur.nl/bioinformatics/iPRESTO/.

**Funding:** JJRL, MHM, and JJJvdH are grateful to
the Netherlands eScience Center for financial
support (ASDI eScience grant, ASDI.2017.030, and
an Open eScience Call, NLESC.OEC.2021.002).
JJRL received a salary from the Netherlands
eScience Center (ASDI eScience grant,
ASDI.2017.030). We note that SvdB (Netherlands
eScience Center) advised with his expertise advised
from his personal views on the software
implementation and validation. As such, the
funders had no role in study design, data collection
and analysis, decision to publish, or preparation of
the manuscript.

**Competing interests:** We have read the journal's
policy and the authors of this manuscript have the
following competing interests: M.H.M. is on the
scientific advisory board of Hexagon Bio and co-
founder of Design Pharmaceuticals. JJJvdH is a
member of the Scientific Advisory Board of
NAICONS Srl., Milano, Italy. All other authors have
declared that no competing interests exist.

## Author summary

In this work, we introduce iPRESTO, a tool for scalable unsupervised sub-cluster pre-
diction in biosynthetic gene clusters. This computational genomics tool development is
important because these biosynthetic hotspots encode many products useful for human-
ity, such as antibiotics, antitumor agents, or herbicides. Recent technological develop-
ments have made detection of biosynthetic loci in genomes straightforward. Yet,
methods to connect these inferred biosynthetic genes to the final chemical structures of
their cognate metabolites are largely lacking. Being able to reliably predict parts of the
final product would constitute a real step forward in natural product genome mining
through integrative omics mining. Therefore, we focussed on constructing a tool to sys-
tematically predict and annotate small regions called sub-clusters, which code for the
biosynthesis of substructures in the final product, across all genomically inferred bio-
synthetic diversity. iPRESTO now makes it possible to query unknown biosynthetic
regions and infer which substructures are present in their metabolic products. This will
facilitate more effective prioritisation of chemical novelty, as well as linking activities
from bioassays and microbiome-associated phenotypes to the metabolites responsible
for them.

This is a *PLOS Computational Biology* Software paper.

## Introduction

A considerable part of bacterial metabolism is dedicated to the biosynthesis of specialised
metabolites. These natural products (NPs) have many uses as pharmaceuticals, crop protection
agents, and ingredients for foods and cosmetics [1,2]. NPs consist of a spectrum of different
chemical classes, which are often highly complex in structure [3]. Intriguingly, the genes neces-
sary for the biosynthesis of NPs cluster together physically in biosynthetic gene clusters
(BGCs) [4]. The search and discovery of new BGCs accelerates identification of new NPs,
which is especially important in the field of antibiotics, as antibiotic-resistant bacteria are
becoming increasingly prevalent [5].

Due to the growing availability of genomic data, genome mining approaches have become
more and more useful for NP discovery. Currently, multiple algorithms exist that mine bacte-
rial genomes for putative BGCs, such as antiSMASH, ClusterFinder and PRISM [6–8]. These
methods have provided a better understanding of BGC diversity and the evolutionary mecha-
nisms that govern BGC diversity.

Many classes of BGCs display a modular architecture [4]. As such, a BGC can be divided
into multiple modules or sub-clusters, where each sub-cluster is a group of co-evolving
genes responsible for the biosynthesis of a specific chemical moiety in the NP [4,9,10]. Sub-
clusters therefore provide a direct link between the substructures of an NP and its BGC. This
makes information about sub-clusters and the substructures they synthesise highly valuable
for genome-based structure prediction, which would be a great asset for tools like anti-
SMASH. Apart from enhancing structural predictions for existing BGC classes, sub-cluster
knowledge would facilitate predicting novel (partial) structures of currently unclassified
BGCs, such as the thousands of unclassified BGCs with yet unknown products in the anti-
SMASH-DB [11].

Additionally, BGC modularity poses a great opportunity to connect metabolomics experiments to sub-cluster data. Chemical moieties identified from fragments in mass spectrometry (MS) data could be linked to sub-clusters responsible for their synthesis, as part of MS-guided genome mining strategies [10,12,13]. Recent advances in substructure modelling [14] may aid such co-occurrence-based metabologenomic approaches [15] by automating the identification of substructures from MS/MS data.

Recently, Del Carratore et al. [10] introduced an initial method for the prediction of sub-clusters in BGCs. By constructing Clusters of Orthologous Groups (COGs) and by using a statistical approach to group co-occurring COGs in sub-clusters, they were able to detect several experimentally characterised sub-clusters, as well as to discover novel ones. However, COG construction is not very scalable due to the all-vs-all BLAST calculation required. As a result, their analysis was performed on a relatively small dataset that is by now almost a decade old, and the chosen approach is hard to scale up to the massive amounts of genomic data that have become available in recent years. Additionally, the proposed statistical approach greatly overestimates the numbers of sub-clusters. This is due to the presence of redundant BGCs, which leads to artificial sub-clusters spanning entire BGCs, and caused by the inherently nested structure of the sub-clusters, where smaller, less specific sub-clusters are contained in larger, more specific sub-clusters. Apart from (artificially) inflating the number of sub-clusters, nested structures also make it more difficult to find actual biologically meaningful sub-clusters as a result of nested combinations of biological sub-clusters.

Here, we propose an improved scalable method for unsupervised sub-cluster prediction which we called the integrated Prediction and Rigorous Exploration of biosynthetic Sub-clusters Tool (iPRESTO). iPRESTO is scalable to large datasets and takes phylogenetic bias into account by filtering the input in a more advanced way. To predict sub-clusters, iPRESTO uses a statistical approach (PRESTO-STAT) as well as a topic modelling algorithm (PRESTO-TOP). PRESTO-STAT uses the same approach as the method by Del Carratore et al. [10] to find sub-clusters based on genes that co-occur in a statistically significant fashion across a collection of BGCs. We further developed the method by removing part of the nested sub-clusters and collapsing similar sub-clusters into families and clans. PRESTO-TOP is a novel method for sub-cluster prediction based on Latent Dirichlet Allocation (LDA) that learns a set of sub-cluster motifs from a collection of BGCs. As a data source, we used the antiSMASH-DB, which is one of the largest collections of BGCs that currently exists, and which has been scrutinized for underlying genome assembly quality [11]; it contains over 150,000 BGCs from almost 25,000 bacterial species selected to reduce taxonomic bias. These numbers represent a considerable improvement in comparison with the previous method as it contains over ten times as many BGCs, while being less redundant. After applying iPRESTO on this large collection of BGCs, we were able to annotate 45 sub-cluster motifs based on occurrences in known BGCs from the MIBiG reference BGC database [16]. Using these annotated sub-cluster motifs, we zoomed in on relevant sub-clusters, and showed direct usefulness of our method by correctly predicting the BGCs for xenorhabdin and salbostatin biosynthesis (which have been published but were missing from BGC databases) and identifying a candidate BGC for akashin biosynthesis. Finally, as a starting point for the automated connection of BGCs to their NPs, we were able to systematically link sub-clusters to substructures by using a metabologenomic correlation method in a paired-genome-metabolome dataset of 145 actinobacteria.

iPRESTO is available as a command-line tool at https://git.wageningenur.nl/bioinformatics/iPRESTO/. We anticipate that the main use of iPRESTO for the genome-mining community is to query BGCs to the annotated sub-cluster motifs that we made available, and, in doing so, predicting one or more substructures in the biosynthetic products of those BGCs. To make our current analyses most useful, we provide both the annotated sub-clusters

as well as the remaining unannotated sub-clusters together with information such as the biosynthetic classes and taxonomic assignments of the BGCs that are associated with them, so that they can continue to be explored. With new training data, it is of course also possible to generate novel sub-cluster models with iPRESTO.

## Results & discussion

### Overview of iPRESTO

iPRESTO prepares each BGC for sub-cluster prediction by tokenising each gene in a BGC as a combination of Pfam domains (Fig 1 and S1 Fig). If a pair of proteins share the same Pfam domains, this provides an effective indication of (at least distant) sequence similarity, while Pfam detection is highly scalable. As Pfams are quite broad sequence models (which would be a major disadvantage compared to using COGs), we increased the resolution by splitting the 112 most abundant biosynthetic Pfams into a number of subPfams, akin to the implementation in BiG-SLICE [17]. Each subPfams constitutes a narrower domain model that covers a subset of a Pfam's sequence space. We only considered biosynthetic domains (see Methods) to limit the search space and focus solely on finding biosynthetic sub-clusters. With a graph-based filtering step, redundant BGCs are removed, after which iPRESTO predicts sub-clusters using PRESTO-STAT and PRESTO-TOP. PRESTO-STAT is based on the previously published statistical method [10], which we expanded by partly removing nested sub-clusters, collapsing similar sub-clusters into families, and joining similar families into clans.

To extend the toolbox for discovery of sub-clusters with a method that does not produce nested sub-clusters, we introduce PRESTO-TOP as a novel approach for sub-cluster prediction. PRESTO-TOP is built on Latent Dirichlet Allocation (LDA), which is used to model topics in text documents. LDA has already been used successfully in genome and metabolome data analysis before [14,18]. In the case of PRESTO-TOP, a text document is a BGC, a word is a gene represented as a domain combination, and a topic can be thought of as a sub-cluster motif. This makes the use of PRESTO-TOP for sub-cluster prediction intuitive, as we assume that a BGC is a combination of multiple different sub-clusters, which consist of co-evolving

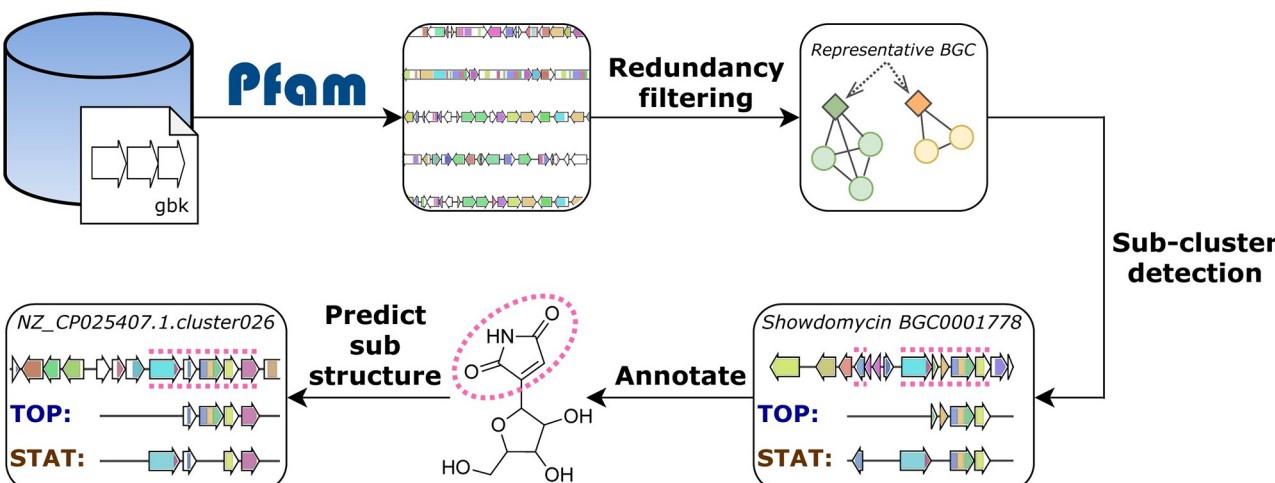

**Fig 1. Outline of the iPRESTO workflow for the prediction of sub-clusters.** All genes in BGCs are converted into strings of Pfam domains, after which redundant BGCs are filtered out based on an Adjacency Index of domains. Sub-clusters are predicted using two methods: PRESTO-TOP (TOP) and PRESTO-STAT (STAT). BGCs from the MIBiG database are used to annotate putative sub-clusters with sub-structures. These annotations are used to predict sub-structures in unknown BGCs.

genes that co-occur in multiple BGCs. Another benefit of PRESTO-TOP is that a topic or sub-cluster motif will usually consist of a set of core genes that encode the enzymes to synthesise the base of a substructure, while various combinations of additional modifying genes can be found in PRESTO-STAT-predicted (nested) sub-clusters. In this way, the two iPRESTO methods can jointly capture substructure diversity, by identifying the sub-cluster cores as well as their variants.

The resulting sub-clusters of both methods can be annotated with substructures and subsequently be used to predict sub-structures in BGCs. iPRESTO is readily usable as a command-line tool for anyone who wants to predict sub-clusters in their own datasets, by querying BGCs to the collection of sub-clusters we predicted and partially annotated in this study. It is also possible to use iPRESTO for predicting new sub-cluster models from scratch using new training data. iPRESTO can handle large amounts of BGCs: tokenising and reducing redundancy in the 150,000 BGCs in the antiSMASH-DB dataset took around 48 hours each using 32 CPU cores on an Intel Xeon CPU E5-2670 v3. Predicting sub-clusters with PRESTO-STAT and PRESTO-TOP completed in 24 and 8 hours, respectively. iPRESTO can query around 20 BGCs per minute to the sub-clusters predicted in this study including the tokenisation steps. iPRESTO also contains a visualisation module to visualise the results of querying a BGC to PRESTO-STAT or PRESTO-TOP output (see S2 Fig for an example of querying the rifamycin BGC).

## PRESTO-STAT improves comprehensibility of existing statistical method

We applied iPRESTO to the antiSMASH-DB v2 dataset, which contained, after pre-processing, 60,028 BGCs with 10,539 domain combinations (Table A in S1 Text). Using the PRESTO-STAT method, we found 108,085 sub-clusters in the dataset. Over 80% of the statistical sub-clusters contain fewer than ten genes, and 17% of the sub-clusters occur in more than 10 BGCs (S3 Fig). When comparing PRESTO-STAT with the previous version of the method by Del Carratore et al. [10], we observed that PRESTO-STAT produces on average roughly two sub-clusters per BGC, while the previous method resulted in roughly fourteen sub-clusters per BGC. This indicates that we end up with fewer nested sub-cluster structures, which is most likely due to our extended redundancy filtering that removed almost half of the dataset (Table A in S1 Text). Even so, nested structures are still very apparent in our results (S2 Fig). For example, thousands of BGCs have more than 30 sub-clusters, many of which overlap with one another (S4A Fig). Not only do the nested structures inflate the results, but they also have the additional disadvantage that their presence makes it harder to connect BGCs with similar yet distinct sub-clusters.

To facilitate the sub-cluster analysis, we connected related sub-clusters by clustering the statistical sub-clusters into 10,000 sub-cluster families (SCFs) and the SCFs into 2,000 sub-cluster clans (SCCs). We used K-means clustering and represented the statistical sub-clusters as a presence/absence matrix of the tokenised genes. Although some SCCs grouped seemingly unrelated sub-clusters together that share only one gene (based on having the same Pfam domain content), most SCCs (81%) provided groups of related sub-clusters, sharing at least three genes.

Apart from the nested structures, the statistical method produces many sub-clusters of which only a fraction probably provides meaningful information. This is illustrated by the fact that the PRESTO-STAT results can be very noisy: in a group of BGCs sharing multiple sub-clusters, all combinations of these shared sub-clusters could form new sub-clusters, which happens frequently (S2 Fig). Additionally, it would quickly become very time-consuming to query a BGC using the statistical sub-clusters while also allowing inexact matching.

## PRESTO-TOP identifies characterised and novel sub-clusters

The drawbacks of PRESTO-STAT present a clear reason as to why we chose to also develop PRESTO-TOP, which can find multiple sub-clusters in a BGC and is able to capture sub-cluster diversity within sub-cluster motifs. Furthermore, LDA, upon which PRESTO-TOP is built, allows for a scalable way to build and query sub-cluster motifs.

We used PRESTO-TOP to train and query a model on the antiSMASH-DB dataset with 1,000 sub-cluster motifs. In the Methods section, we provide information on (hyper)parameters used and reasoning for the chosen settings. Over 80% of the BGCs in the dataset contained at least one sub-cluster motif (S4B Fig). To assess the quality of the sub-cluster motifs, we visualised all sub-clusters individually, where each sub-cluster is a group of genes matching against a sub-cluster motif (Fig 2A). For a sub-cluster to be interesting, we would expect its size to be between 2–12 genes, as experimentally characterised sub-clusters fall in this range [19]. Upon checking our results, most sub-clusters that were present across a considerable number of BGCs were within this expected size range (Fig 2A), while some sub-clusters were uninformative as they encompass (nearly) entire BGCs (Fig 2B). To validate the sub-cluster motifs, we assessed whether we could detect a set of 109 experimentally verified sub-clusters, which are stored in the SubClusterBlast module within the antiSMASH framework. The sub-cluster motifs from PRESTO-TOP matched to 91 (83%) validated sub-clusters, where the methoxy-malonate and AHBA sub-clusters of macbecin are shown as examples (Fig 2C). Additionally, PRESTO-STAT was able to detect 78 of the validated sub-clusters, of which 75 overlap with the sub-cluster motifs (S5 Fig). In general, we see that PRESTO-TOP generates a more restricted amount of sub-cluster data, which might contain less meaningful sub-clusters compared to PRESTO-STAT in absolute numbers but has a considerably higher ratio of valid sub-cluster information.

Our results provide clear examples of sub-cluster motifs that capture sub-cluster variety, by containing a set of core genes responsible for synthesising the base of a substructure, and a set of modifying genes that may not be present in all sub-clusters. For example, a motif like the

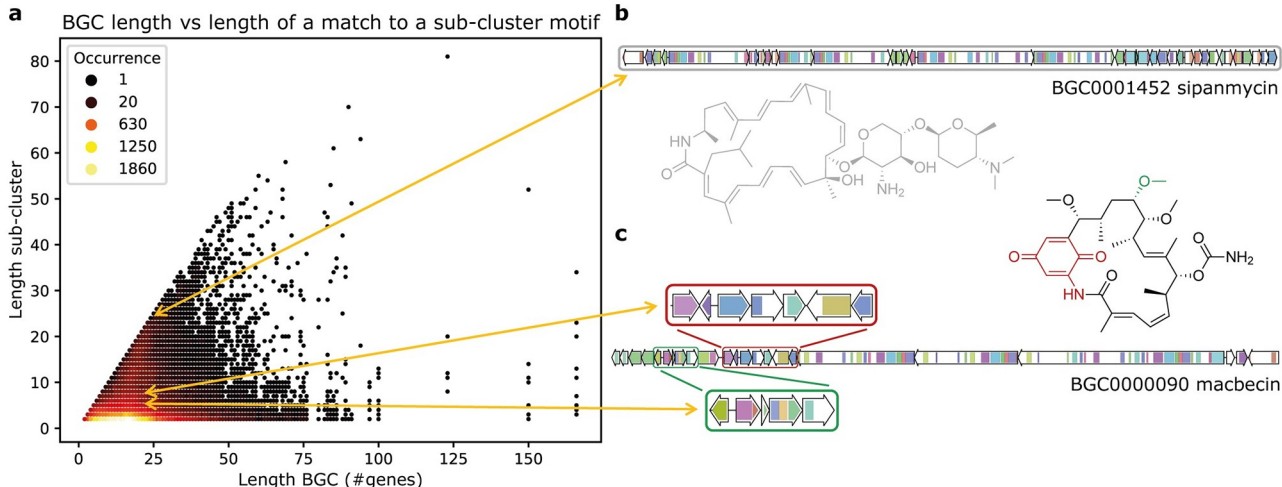

**Fig 2. BGC length versus sub-cluster length.** (a) Scatterplot of the length of each BGC (number of non-empty genes) from the antiSMASH-DB dataset versus the length of a match to a topic or sub-cluster motif, representing a sub-cluster. The colour of each dot indicates how many times a BGC with a certain length contains a sub-cluster with a certain length. (b) BGC for sipanmycin where the identified sub-cluster encompasses the entire BGC, demonstrating an uninformative result. (c) BGC for macbecin where the two characterised sub-clusters for AHBA (red) and methoxymalonyl (blue) are highlighted in the structure of macbecin [20]. Sub-clusters from (b) and (c) are linked to their corresponding location in (a).

sugar-related sub-cluster motif 680 is present in 134 MIBiG BGCs that represent different biosynthetic classes, such as different types of polyketide synthases and nonribosomal peptide synthetases. This motif codes for the biosynthesis of different (di)deoxy-sugars that are sometimes modified with amino or methyl-amino groups. However, for some sub-cluster motifs, the biosynthetic context had an impact on shaping the motif. The sugar-related sub-cluster motif 207, for example, contains several indolocarbazole biosynthesis genes as some MIBiG BGCs matching to this motif encode the production of indolocarbazoles, and some of the indolocarbazole-related genes ended up in this motif as weak features.

## Exploring the sub-cluster motifs

Among the 90 identified characterised sub-clusters from the antiSMASH SubClusterBlast module, we could readily annotate 23 sub-cluster motifs covering around 4,000 of the PRESTO-TOP-predicted sub-clusters. To extend on the sub-cluster knowledge stored in the SubClusterBlast module, we annotated another 22 PRESTO-TOP-predicted sub-cluster motifs for which sub-cluster instances were found inside MIBiG BGCs. Together, these 45 annotations constitute 24 different types of substructures at different levels of detail and allow us to explore the discovered sub-clusters more deeply (Fig 3 and S1 File). In the non-redundant antiSMASH-DB dataset, around 9,500 (16%) putative BGCs contain at least one of these annotated sub-cluster motifs. Through iPRESTO, we now gained relevant knowledge about these putative BGCs that we can use to predict part of the structures of the products they encode.

On average, an annotated sub-cluster motif occurs in 239 non-redundant BGCs, ranging from 19 BGCs for sub-cluster motif 190, to 873 BGCs for sub-cluster motif 220, which encode

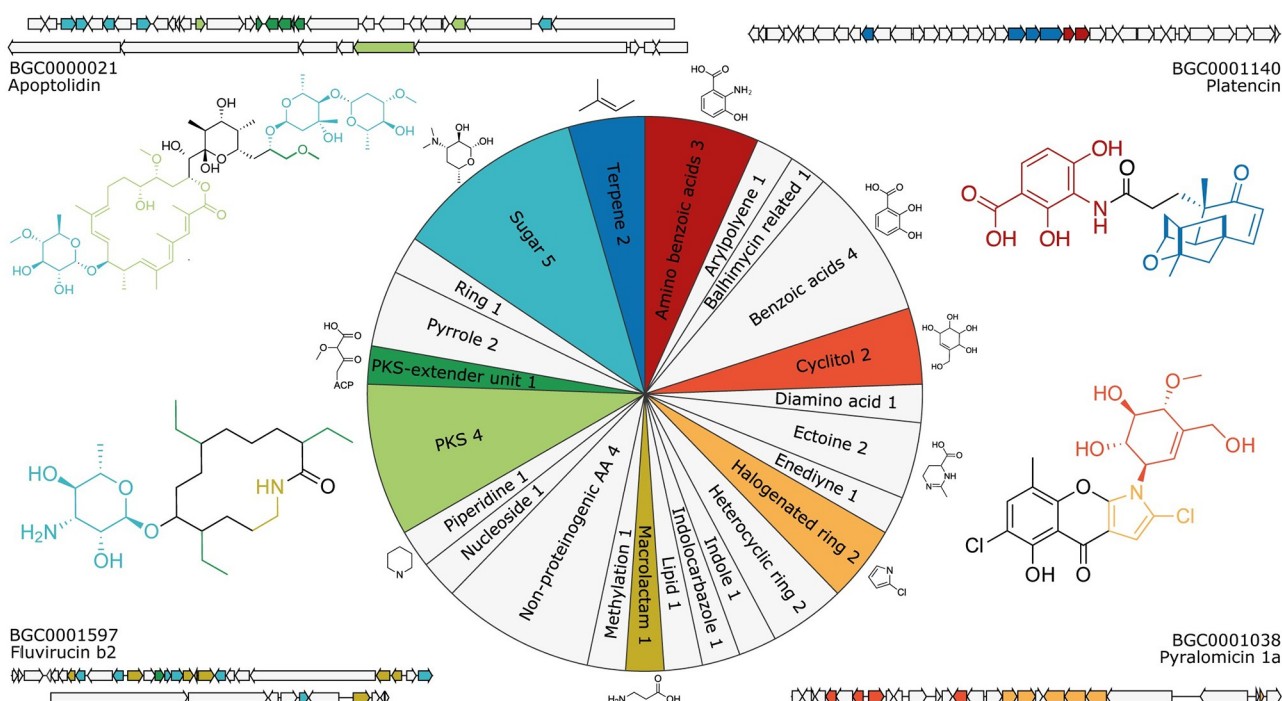

**Fig 3. Sub-cluster motif annotations.** The pie chart visualises the annotations for the 45 sub-cluster motifs divided into general substructure groups, where an example substructure is shown for several groups. Additionally, examples of eight of the substructures are shown in the structures of apoptolidin, platencin, fluvirucin b2 and pyralomicin 1a, where the colours of the substructures correspond to the sub-cluster motif annotations in the pie chart. For these four metabolites, their respective BGCs are shown where the sub-cluster motifs are highlighted in the same colour as the substructures they encode.

the biosynthesis of caprazol and dihydroxybenzoic acid moieties, respectively (S6 Fig). Some of the annotated sub-cluster motifs are mainly present in one BGC class, while others occur in diverse BGC classes (S6 and S7 Figs). An example of the latter is sub-cluster motif 773, which occurs in 153 BGCs mostly encoding nonribosomal peptide synthetases and type I polyketide synthases. This sub-cluster motif encodes the production of a 3-amino-2-methylpropionyl starter unit that appears in the known gene cluster BGC0001597 (fluvirucin b2) (Fig 3). Interestingly, the motif also occurs in some BGCs of the class "Other", meaning they cannot be classified by antiSMASH, like two BGCs from *Amycolatopsis alba* DSM 44262 (NZ_KB913032.1. cluster021; AMYAL_RS0129245—AMYAL_RS0129610) and *Bradyrhizobium sp*. *Ec3.3* (NZ_AXAS01000001.cluster006; YUU_RS0100020—YUU_RS49645). This does not only provide interesting leads for these BGCs with previously unknown structural predictions, but it also adds to their validity. In total, 6.5% of the 10,000 "Other" class BGCs in the anti-SMASH-DB contain one of the annotated sub-cluster motifs.

## iPRESTO can identify BGCs of orphan metabolites through sub-cluster presence

Information about the sub-clusters present in a BGC is not only useful to predict the product of a BGC, but it could also be used as a tool to identify BGCs for 'orphan' known metabolites. To demonstrate this, we searched NPAtlas [21] with substructures that are encoded by our annotated sub-cluster motifs and looked for metabolites without a MIBiG BGC that are found in one of the strains in the antiSMASH-DB dataset. We first searched for metabolites that contain the dithiolopyrrolone substructure for which the biosynthesis is encoded by sub-cluster motif 517, annotated as such based on the MIBiG BGCs encoding thiomarinol, holomycin and thiolutin [22–24]. In doing so, we found xenorhabdins 1–6, produced by many *Xenorhabdus* strains that are also present in the antiSMASH-DB [25]. By searching for BGCs in those strains that contain a match to the dithiolopyrrolone sub-cluster motif, we found 12 *Xenorhabdus* strains that contain such a BGC (Fig 4). In one of those strains, *X. doucetiae*, the BGC for xenorhabdin biosynthesis has recently been described, corroborating that we accurately identified BGCs for xenorhabdin biosynthesis based on iPRESTO-predicted sub-clusters [26]. Next, we searched NPAtlas for metabolites with the valienol moiety present in validamycin and pyralomicins, which is encoded by sub-cluster motif 940 [27,28]. As a result, we found salbostatin, which is produced by *Streptomyces albus* ATCC 21838 in our dataset [29]. By investigating BGCs in that strain, we identified a BGC that contains sub-cluster motif 940 and should therefore be responsible for salbostatin biosynthesis (Fig 4). Indeed, it turned out that this BGC has already been described in 2008 to encode the production of salbostatin [30], but it has been lacking from the MIBiG database [16]. This valienol sub-cluster motif encoding C7-cyclitol-like substructures is an interesting example of a sub-cluster motif that can be found in different biosynthetic contexts, *i.e.*, PKS-NRPS-like pyralomicins and different kinds of saccharides like validomycin and salbostatin. This analysis highlights that iPRESTO allows identifying correct links between BGCs and molecules that are published but were yet missing in public BGC databases (and which can thus be added to these resources).

By searching in NPAtlas for chlorinated indoles, we found the orphan metabolites akashin A-C produced by the diazaquinomycins producer *Streptomyces* sp. F001 [32]. The BGC of akashins has not been described before in literature. As this strain was not present in the anti-SMASH-DB, we ran antiSMASH 6 on the genome of this strain and used iPRESTO to infer sub-clusters in the predicted BGCs. As akashins have chlorinated-indole moieties and are glycosylated, we sought for such sub-cluster motifs in the BGCs of *S*. sp. F001. Interestingly, we identified the genomic region in QZWF01000007.1.region003 (StrepF001_25985—

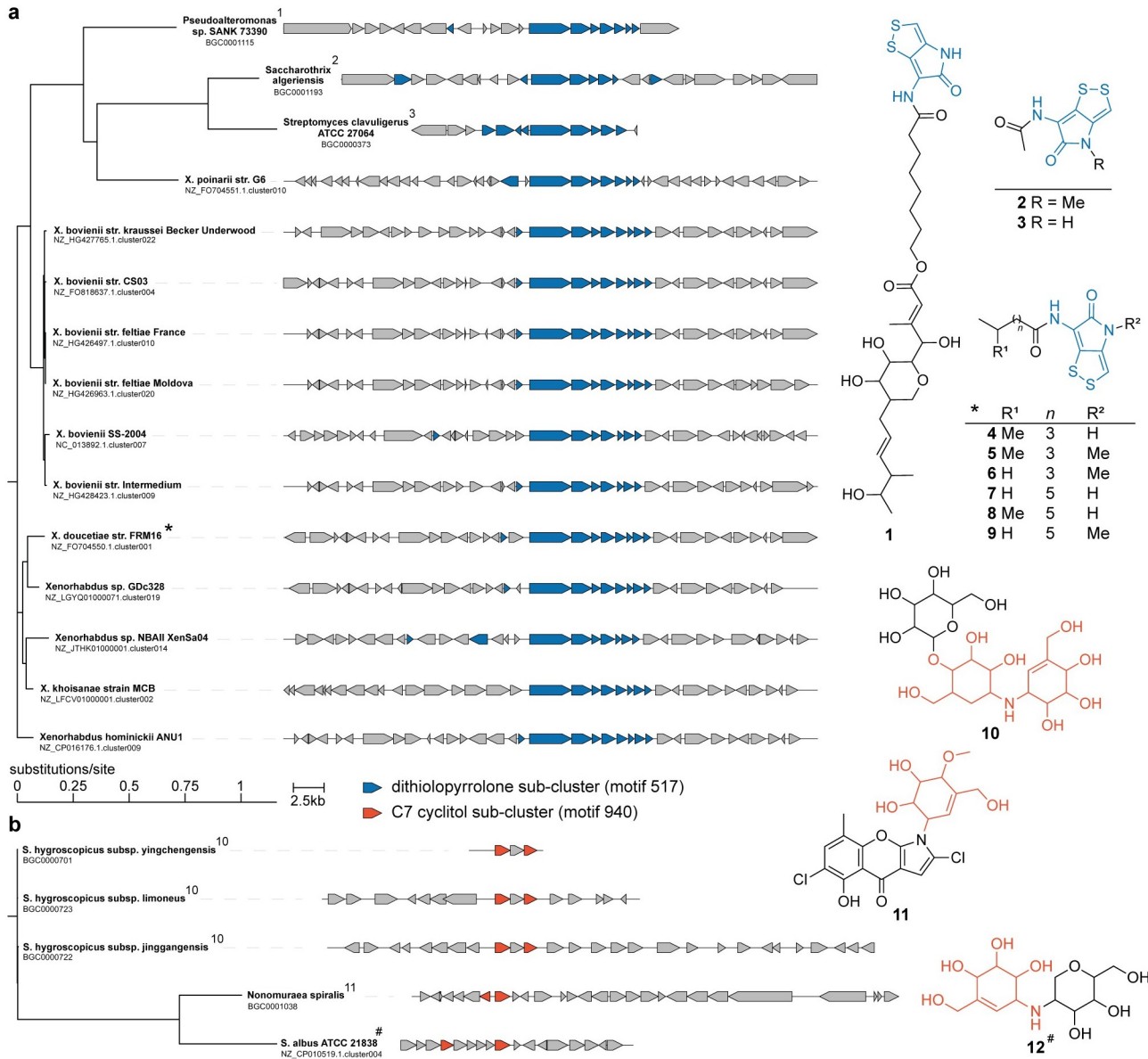

**Fig 4. Connecting non-MIBiG BGCs to their metabolic products through iPRESTO-predicted sub-clusters.** (a) Phylogenetic tree made with CORASON of 12 *Xenorhabdus* BGCs and 3 MIBiG BGCs, that contain an iPRESTO-predicted sub-cluster for dithiolopyrrolone biosynthesis [31]. The A-domain containing gene of NZ_FO704550.1.cluster001 was used as query for CORASON. Structures of thiomarinol (1), thiolutin (2) and holomycin (3) are linked to their MIBiG BGCs. Xenorhabdins (4–9) are encoded by *X. doucetiae* str. FRM16 as indicated by the asterisk, while we infer based on sub-cluster presence that the other *Xenorhabdus* BGCs are also responsible for xenorhabdin biosynthesis. (b) Phylogenetic tree made with CORASON NZ_CP010519.1.cluster004 from *S. albus* ATCC 21838 and 4 MIBiG BGCs, that contain an iPRESTO-predicted sub-cluster for C7 cyclitol biosynthesis. The predicted 2-epi-5-epi-valiolone synthase from NZ_CP010519.1.cluster004 was used as query for CORASON. Structures of validomycin A (10) and pyralomycin 1A (11) are linked to their MIBiG BGCs. Salbostatin (12) is encoded by *S. albus* ATCC 21838 as indicated by the hash symbol.

StrepF001_26130) directly upstream of the diazaquinomycin BGC, based on the presence of sub-cluster motifs 194, 607 and 680 that were annotated as methylaminosugar, halogenated aromatic ring, and (amino)deoxysugar, respectively (Fig 5). The formation of the indigo-derived backbone of akashins could potentially be formed by the two p450 enzymes, akin to CYP102G4, a recently described p450 enzyme from *S. cattleya* [33]. This p450 enzyme can catalyse the reaction from indole to 3-hydroxyindole after which spontaneous oxidation forms

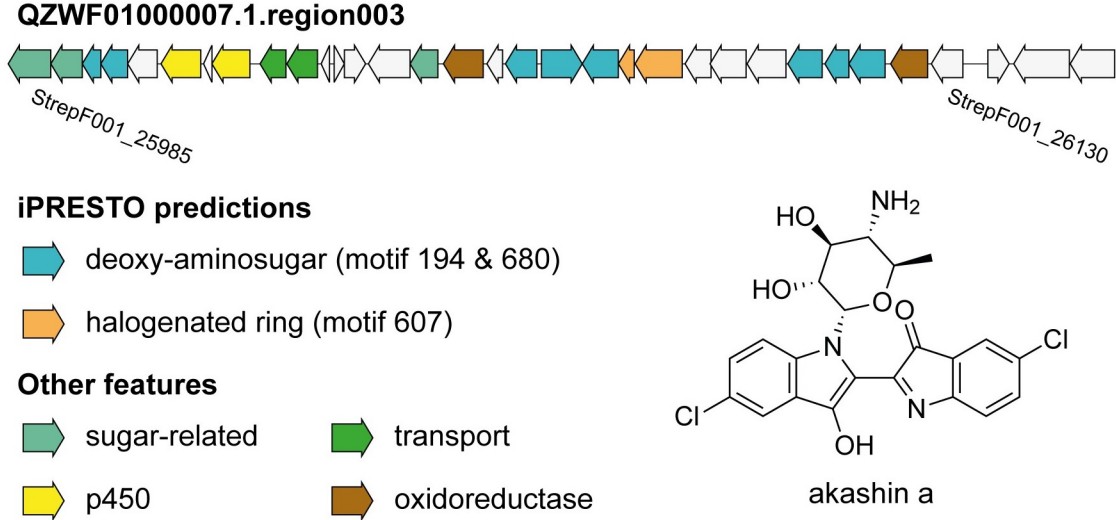

**Fig 5. Putative BGC for akashin A biosynthesis.** The antiSMASH-predicted BGC QZWF01000007.1.region003 is shown (StrepF001_26130-StrepF001_26145), which is hypothetically responsible for akashin A biosynthesis in *S.* sp. F001. Genes are coloured by their iPRESTO-predicted sub-clusters or predicted function based on Pfam domains.

indigo. CYP102G4 was even shown to accept chloro-indole as substrate, in the case that chlorination occurs before indole formation in akashin biosynthesis. This shows that iPRESTO can aid in generating meaningful hypotheses about the biosynthesis of orphan metabolites.

## Correlation analysis in substructure-based integrative omics mining

To automatically link unknown molecules to BGCs at a larger scale, correlating substructures predicted from metabolomics data to sub-clusters from genome data would potentially be of great added value [12,13]. To test such an approach, we used a previously defined correlation score which assumes that a BGC is needed to synthesise a product, but that a BGC may be cryptic and not synthesise anything under the used conditions [15]. Ernst et al. [34] used the MS2LDA tool to discover substructure mass patterns, called Mass2Motifs, from metabolomics data of 145 *Salinispora* and *Streptomyces* species for all of which (except one) genomic data and BGC predictions are also available (the 'Streptomyces/Salinispora dataset') [14]. To identify sub-clusters in the genomics data of the same species, we used iPRESTO to query all Streptomyces/Salinispora BGCs on the sub-cluster motifs and sub-cluster clans (SCCs) of the antiSMASH-DB dataset. For each of the 107,590 pairs of Mass2Motifs and sub-cluster motifs, we used the correlation score from Doroghazi et al. [15] to calculate how frequently they co-occur across the Streptomyces/Salinispora strains, while we did the same for the 122,404 pairs of Mass2Motifs and SCCs (S8 Fig). To prioritise interesting substructure-sub-cluster pairs, we performed permutation tests for all pairs to assess the likelihood of a high scoring pair arising by chance. This was especially needed as the Streptomyces/Salinispora dataset includes highly related strains, in which many BGCs and compounds are shared. Abundant sub-clusters and substructures therefore get high correlation scores by default. Permutation testing resulted in 3,230 and 1,939 'significant' pairs of Mass2Motifs and sub-cluster motifs or SCCs, respectively (S8 Fig). As an example of how such an approach connects substructure information inferred from genome mining with that of metabolome mining, we identified 5 high correlation scores with low p-values between two staurosporine-related mass2motifs and both sub-cluster motifs and SCCs constituting the amino-sugar moiety of staurosporine (Fig 6). Since currently only a

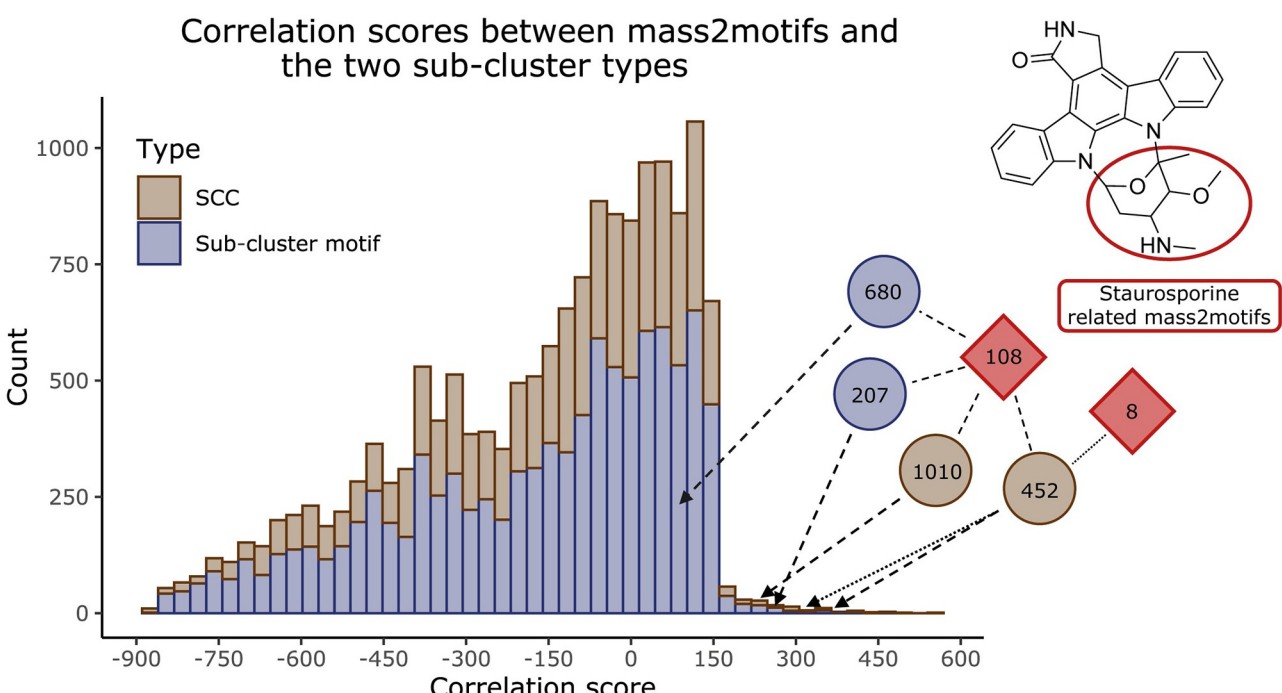

**Fig 6. Metabologenomic correlation scores between sub-clusters and mass2motifs.** Stacked histogram of the correlation scores across the Streptomyces/Salinispora strains between the mass2motifs paired with either the SCCs or sub-cluster motifs with a p-value below 0.1. Highlighted with their scores are the pairs mass2motif_108 with SSC_452, SSC_1010, sub-cluster_motif_207 and sub-cluster_motif_680, and the pair mass2motif_8 with SSC_452. The aforementioned sub-cluster motifs (blue) and SCCs (brown) are responsible for sugar synthesis in staurosporine, while both mass2motifs (red) are staurosporine related.

fraction of the Mass2Motifs, sub-cluster motifs and SCCs are annotated, our analysis serves as an illustration of how such an approach could help to link metabolome and genome data in the future.

This correlation method generally results in a lot of noise, as sub-clusters and substructures that occur in a shared subset of strains will all correlate to each other. Therefore, such co-correlating structures make the identification of the actual correlating pair difficult, especially with limited annotations. Identifying clusters of co-correlating pairs could provide a way to make the interpretation of this analysis easier. Additionally, the correlation analysis is not perfect in our case, as multiple different sub-clusters are often responsible for synthesising the same kind of substructure. For example, we identified multiple sub-cluster motifs that can encode for the production of methylated aminosugars, while only one mass2motif is annotated as a methylated aminosugar. In future approaches, such mismatches between genome and metabolome could be overcome by finding ways to group sub-cluster motifs together that encode similar structures before running such metabologenomic correlation analyses. Combining such solutions with the integration of more diverse species, new annotations, and improved correlation scoring methods like the one developed in Hjörleifsson Eldjárn et al. [35] would improve such analyses drastically. Furthermore, we expect that combining co-occurrence based scores (such as the standardised Metcalf one) with feature-based scores, such as NPClassScore [36], and the here developed iPRESTO, will further help to prioritise plausible BGC-MS/MS spectral links [12,13]. Indeed, we expect that tools like iPRESTO could in the future be built into frameworks like NPLinker [35]. As our current contribution represents a first step in linking substructure-and sub-cluster models with rather limited (annotated)

information, we expect that analyses like these will have great impact in the future to facilitate metabologenomics experiments that use integrative omics mining.

## Conclusion and future perspectives

This study introduces the iPRESTO concept and makes it available as a command- line tool which can be used to query BGCs to the set of partially annotated sub-clusters generated in this study, as well as to train new sub-cluster models. We plan to include iPRESTO in one of the future releases of antiSMASH, so the collection of sub-clusters we generated can be used more easily to predict and visualize them in antiSMASH-predicted BGCs. We anticipate that this will enhance the current scope of sub-cluster prediction, as antiSMASH's current sub-clus-ter predictor SubClusterBlast offers a limited amount of sub-cluster data, whereas our sub-cluster set will allow making more connections between predicted BGCs and MIBiG reference BGCs. This will accelerate NP discovery by linking structural information from genome and metabolome data.

Due to the above discussed limitations of PRESTO-STAT, we plan to use PRESTO-TOP as the main method for sub-cluster prediction in the antiSMASH implementation, as it also cap-tures sub-cluster variety in the sub-cluster motifs and yet can be used easily to query BGCs for sub-cluster motifs. PRESTO-STAT could still be used to identify the sub-cluster boundaries better, by for example linking groups of related PRESTO-STAT sub-clusters to 'parent' PRE-STO-TOP sub-cluster motifs, and by using the PRESTO-STAT modules to more specifically identify the sub-cluster variant found in a given BGC. The drawback of the statistical method, i.e., that it produces highly nested and variable sub-clusters, could as such be used as a strength. A way to further improve PRESTO-TOP would be to apply PRESTO-TOP in a semi-super-vised manner, which constitutes a major potential benefit of this approach. Before training an LDA model, certain motifs could be seeded beforehand, which allows accurate sub-cluster motifs to be reused in new analyses, analogous to the metabolomics substructure database MotifDB, in which annotated Mass2Motifs are stored in MotifSets [37]. Such semi-supervised approaches would allow for noise to be eliminated from sub-cluster motifs and sub-cluster motifs to be finetuned. Another way to reduce noise and to identify the more robust sub-clus-ter motifs would be to train multiple PRESTO-TOP models on the same dataset. Sub-cluster motifs that are found in every PRESTO-TOP model would constitute conclusive sub-cluster motifs, whereas sub-cluster motifs that are identified in most cases would still be considered reasonably accurate. In this manner, noisy sub-cluster motifs that arise by chance would be fil-tered out, as they would only occur in one or a few of the many LDA models. Noisy genes in accurate sub-cluster motifs could be filtered out by taking the intersection of multiple similar sub-cluster motifs. As another option, each BGC could be represented multiple times in train-ing to increase the observations of less frequently occurring sub-clusters. This could lead to better estimation of the sub-cluster motif distributions over the data and cause less erroneous mixed sub-cluster motifs. We have attempted this for a small subset and noticed that the over-lap with SubClusterBlast increased slightly, making this an interesting avenue to continue PRESTO-TOP sub-cluster algorithmic developments.

Using iPRESTO, in our current study we were able to characterise 45 different sub-cluster motifs present in diverse BGC classes. The remaining 955 sub-cluster motifs remain largely unexplored, of which many are likely to encode useful substructures. We expect that, in the future, more annotations will increase the value of our results even more, which will be aided by the inclusion of updated (expanded) versions of the MIBiG database. Using one of the char-acterised sub-cluster motifs, we showed a direct practical application of our method by hypothesising a putative BGC for akashin A production. Additionally, we provided the initial

step for linking genomics-derived sub-clusters to metabolomics-derived substructures in a systematic way, which in the future could facilitate the automated connection of BGCs to their NPs through integrative omics mining.

## Methods

### Data selection

The antiSMASH-DB dataset consisted of three data sources: the MIBiG database, the Streptomyces/Salinispora dataset and the antiSMASH-DB. Version 1.4 of the MIBiG database was used which contains 1,819 BGCs (https://dl.secondarymetabolites.org/mibig/mibig_gbk_1.4.tar.gz). The Streptomyces/Salinispora dataset consists of 5,927 BGCs that originate from the 146 *Streptomyces* and *Salinispora* strains investigated by Crüsemann et al. [38]. antiSMASH 3.0 was used for identification of BGCs in the Streptomyces/Salinispora dataset. The antiSMASH-DB version 2 is comprised of 152,122 BGCs detected with antiSMASH 4.0, where we included BGCs from draft genomes (Table A in S1 Text; https://dl.secondarymetabolites.org/database/2.0/asdb_20180828_all_results.tar.xz). BGCs were discarded if they were flagged by antiSMASH as lying on a contig-edge, as these BGCs are probably incomplete (fragmented) and less accurate. Additionally, BGC class information was included in the analysis, by using the assigned antiSMASH biosynthetic classes.

### Data pre-processing

BGCs were tokenised by converting each gene into a string of (sub)Pfam domains. To identify (sub)Pfams, the HMMER3 tool hmmscan was used with a custom profile hidden Markov model (pHMM) database consisting of Pfam database version 32.0, where 112 Pfams were replaced by corresponding subPfams [39,40]. These 112 Pfams were selected as they are the most abundant biosynthetic Pfams in the antiSMASH-DB (S2 File). To create subPfams, the multiple sequence alignment of a Pfam is split into clades, after which a new pHMM is built for each clade, each of which constitutes a subPfam (S1A Fig and https://github.com/satriaphd/build_subpfam).

Redundant BGCs were removed from the analysis using a similarity network of BGCs, where BGCs were connected based on an Adjacency Index of domains higher than 0.95 or if BGCs were fully contained within one another. From each maximal clique in the network, only the BGC with the most domains was chosen to remain in the analysis (Table A in S1 Text and S9 Fig) [41]. After redundancy filtering, all non-biosynthetic domains were removed from all BGCs. To select biosynthetic domains, EC-associated Pfams were collected with ECDomainMiner, from which Pfams were selected if they occurred in pre-calculated BGCs [42]. After manual curation, this resulted in a list of 1,839 biosynthetic Pfams (S3 File). Additionally, Pfams that occurred less than three times in the dataset were removed as well as BGCs that contained less than two non-empty genes (S4 File).

### PRESTO-STAT

The statistical method for sub-cluster prediction was re-implemented in Python based on Del Carratore et al. [10] with some alterations, resulting in PRESTO-STAT. Instead of representing genes as COGs as in the previous method, we represent each gene as a combination of its domains. First, all possible adjacency and co-localisation interactions between each pair of genes are counted. To assess whether an observed interaction between two genes occurs more than by random chance, one needs to distribute such a pair of genes randomly through the dataset and calculate the probability of the observed interaction. To reduce the computational

burden of a permutation-based approach, for each pair of genes one gene is kept fixed while the other is being randomly distributed throughout the data. For an adjacency interaction this gives a hypergeometric equation describing all available positions of one gene while the other is fixed (Table B1 in S1 Text). This follows from the fact that there are three options for the position of gene B while keeping gene A fixed: not adjacent to gene A ($B_1$), adjacent to gene A ($B_2$), or adjacent to gene A on both sides ($B_3$). $N_1$, $N_2$ and $N_3$ represent all available positions in these three categories, while $N_{tot}$ represents all positions and $B_{tot}$ all occurrences of gene B. For a co-localisation interaction the same applies, except for the fact that gene B can be co-localised with $n_{max}$ genes A, where $n_{max}$ is the number of genes A co-localised with gene B (Table B2 in S1 Text). When $n_{max}$ is large this becomes computationally hard, which is why we replaced duplicate genes with an empty gene (a dash) and placed one copy of the duplicate gene at the end of the cluster separated by an empty gene. This simplifies the equation, as only two types of co-localisations need to be counted: co-localisation and no co-localisation (Table B3 in S1 Text). A p-value can be calculated by summing all probabilities in the hyper-geometric distribution that correspond to several interactions higher or equal to the observed number of interactions. Or, to make it easier, by subtracting the sum of all possible interactions smaller than the observed interaction from one (Table B4 in S1 Text).

Calculating an interaction between each pair of genes results in two p-values, one coming from gene A and one coming from gene B. Only the largest p-value for both the co-localisation, and the adjacency interactions is considered, to be conservative. To control false discovery rate under dependency we used the Benjamini–Yekutieli method on both the co-localisation and adjacency p-values [43].

To group interacting pairs of genes into sub-clusters, undirected graphs are constructed, where each gene is a node. An edge is made between two genes if they have an adjacency or co-localisation p-value below a threshold of 0.1. All maximal cliques are selected as sub-clusters, while changing the threshold iteratively to all the p-values in the dataset smaller than the original threshold of 0.1. To reduce false positives, we removed putative sub-clusters if they contained fewer than three genes and if they only occurred in one BGC. Next, we grouped similar sub-clusters together using K-means clustering into sub-cluster families and sub-cluster clans and removed redundant sub-clusters (Supplementary methods in S1 Text) [44,45].

## PRESTO-TOP

PRESTO-TOP uses Latent Dirichlet Allocation (LDA) latent sub-cluster composition in BGCs [46]. LDA assumes a bag-of-words representation, where each BGC is depicted as a frequency vector of its domain combinations, not taking gene order into account. We used the multicore LDA implementation from Gensim, that makes use of online variational Bayes [47,48]. In this implementation, an LDA model is trained by updating it with mini-batches from the data, which has low time and memory complexity. We chose the chunk size of each mini-batch to be 5% of the data with a minimum chunk size of 2,000, which is loosely based on testing different chunk sizes by Hoffman et al. [48]. We considered that using 500 iterations to train a model was enough after assessing that the log-likelihood converged sufficiently (S10 Fig). For the sake of computational resources, we did limited hyperparameter optimisation for the number of sub-cluster motifs (topics) N, α, and β. To test the performance of the different models, we considered the coherence score as measured with the u_mass method [49] and the overlap with validated sub-clusters from SubClusterBlast (Supplementary methods in S1 Text). Based on the coherence score of the different models, choosing 250 sub-cluster motifs seemed optimal (S11A Fig). However, upon manual

inspection of some of the motifs, it turned out that many motifs are hard to annotate with a single substructure due to the presence of many noisy features. This is corroborated by the fact that choosing 250 sub-cluster motifs does not produce the highest overlap with SubClusterBlast (S11B Fig). Instead, the model with 1000 sub-cluster motifs produced the highest overlap with SubClusterBlast while having a similar coherence score to the model with 250 motifs, which is why we chose 1000 sub-cluster motifs. We chose the default setting of a symmetric 1/N for hyperparameters $\alpha$ and $\beta$, as we could not find a better SubClusterBlast overlap when setting $\alpha$ and $\beta$ to symmetric, asymmetric, auto, or 1.

Each sub-cluster motif in an LDA model consists of a probability vector of domain combinations, representing the contribution of each domain combination to a sub-cluster motif. To filter out noise, we sorted this vector from high to low probability, summed the probabilities and included all domain combinations until 0.95 was reached. When a group of genes from a BGC match to a sub-cluster motif, each gene is assigned a gene-to-motif probability describing how well it fits in the sub-cluster motif, for which we set a cut-off of 0.3. To consider the matching group of genes a sub-cluster, it needs to consist of more than one gene. We therefore set a cut-off of 1.1 on the summed gene-to-motif probabilities. Additionally, we calculated an overlap score for each match, which we computed by summing the domain combination probabilities from the sub-cluster motif present in the match [50]. We set a threshold of 0.15 on the overlap score, as this was the highest threshold that did not remove manually validated SubClusterBlast sub-clusters from the analysis.

## Supporting information

**S1 Text. Supplementary information for iPRESTO: Automated discovery of biosynthetic sub-clusters linked to specific natural product substructures.**
(DOCX)

**S1 Fig. Schematic depiction of BGC tokenisation.** (A) subPfams are constructed for the 112 most frequent Pfam domains in the antiSMASH-DB by dividing the multiple sequence alignment of a Pfam into clades and converting each clade into a new pHMM. (B) The BGCs predicted by antiSMASH are tokenised by detecting (sub)Pfams in each gene, where non-biosynthetic Pfams are removed. After tokenising the BGCs, sub-cluster can be predicted with the statistical method (Stat), where the tokenised genes are represented in their original order, or by LDA, which assumes a bag of words model where original gene order is not considered.
(TIF)

**S2 Fig. Result of querying rifamycin (BGC0000373) to the PRESTO-TOP and PRESTO-STAT sub-clusters generated in this project.** Only around 25% of the PRESTO-STAT sub-clusters are shown. Each gene is depicted as a token, where all (sub)Pfam domains are coloured. The visualisation of the BGC, the PRESTO-TOP and PRESTO-STAT output are separated by a dashed line, respectively. All PRESTO-STAT sub-clusters clearly exhibit a nested structure, where all combinations of genes in an actual sub-cluster are predicted as individual sub-clusters. The PRESTO-STAT sub-clusters shown here are also examples of noisy sub-clusters comprised of combinations of genes from different actual sub-clusters, like predicted PRESTO-STAT sub-clusters that are combinations of genes responsible for the biosynthesis of AHBA (green), sugars (blue) and the polyketide scaffold (purple).
(TIF)

**S3 Fig. Information about the PRESTO-STAT sub-clusters.** (A) The distribution of the number of genes per PRESTO-STAT sub-cluster in the antiSMASH-DB dataset. (B) The distribution of the log10 transformed PRESTO-STAT sub-cluster occurrences in the

antiSMASH-DB dataset.
(TIF)

**S4 Fig. Number of PRESTO-STAT and PRESTO-TOP sub-clusters per BGC.** (A) Distribution of the log10 transformed number of PRESTO-STAT sub-clusters per BGC in the non-redundant antiSMASH-DB dataset, where the bin with the seemingly negative value represents BGCs without any PRESTO-STAT sub-cluster. (B) The number of topics or sub-cluster motifs per BGC in the non-redundant antiSMASH-DB dataset, not counting sub-clusters of length one as these are almost definitely noise (see Methods). (C) All BGCs with at least one annotated sub-cluster motif grouped by how many annotated sub-cluster motifs they have. In total there are 9,425 putative BGCs with at least one annotated sub-cluster motif, and 350 MIBiG BGCs.
(TIF)

**S5 Fig. PRESTO-STAT and PRESTO-TOP overlap with validated sub-clusters from Sub-ClusterBlast.** Overlap between predicted SubClusterBlast sub-clusters and output of both sub-cluster prediction methods applied on the antiSMASH-DB dataset according to different overlap cut-offs. The overlap expresses the fraction of genes from the original SubClusterBlast sub-cluster that is found in the iPRESTO-predicted sub-cluster. We considered an overlap of 0.6 sufficient for having predicted a sub-cluster (see Supplementary methods in S1 Text).
(TIF)

**S6 Fig. Degrees (occurrences) of the annotated sub-cluster motifs within the anti-SMASH-DB dataset (non-redundant).**
(TIF)

**S7 Fig. BGC class distribution across sub-cluster motifs.** Relative abundance of antiSMASH classes when querying the non-redundant antiSMASH-DB dataset on the 45 annotated sub-cluster motifs. Matches of length 1 are ignored and hybrid class BGCs are counted for all classes they contain. RIPPs classes are grouped together.
(TIF)

**S8 Fig. Correlation scores between Mass2Motifs and sub-clusters.** (A) Correlation scores between Mass2Motifs and SCCs. (B) Correlation scores between Mass2Motifs and sub-cluster motifs. In both panels the significant pairs are highlighted.
(TIF)

**S9 Fig. Graphical representation of graph-based filtering for the small dataset: MIBiG-and Streptomyces/Salinispora BGCs.** Each node represents a BGC and an edge represents an adjacency index (AI) of 0.95 or higher. In blue are the BGCs chosen as representatives, while BGCs that are filtered out are shown in black. We show the small dataset here as it was difficult to visualize this process for the antiSMASH-DB dataset.
(TIF)

**S10 Fig. LDA model convergence.** Convergence of the log-likelihood of an LDA model with 1,000 topics/sub-cluster motifs trained on the non-redundant 60,028 BGCs from the anti-SMASH-DB dataset, which also contains the Streptomyces/Salinispora dataset and the MIBiG database, using 2,000 iterations of chunk size 3,000. Log-likelihood based on 28 held out BGCs.
(TIF)

**S11 Fig. Coherence scores and overlap with SubClusterBlast sub-clusters for different LDA models.** (A) Coherence scores of different LDA models trained using PRESTO-TOP

on the non-redundant antiSMASH-DB dataset with different number of topics. (B) Number of validated SubClusterBlast sub-clusters found with different LDA models trained using PRESTO-TOP on the non-redundant antiSMASH-DB dataset with different number of topics.
(TIF)

**S1 File. Excel file containing the current information about the generated sub-cluster motifs.** Sheet one includes annotations for the annotated sub-cluster motifs and sheet two contains metadata for all sub-cluster motifs such as biosynthetic classes and taxonomic assignments of BGCs that are associated with them.
(XLSX)

**S2 File. The 112 domains for which we created subPfams.**
(TXT)

**S3 File. The biosynthetic domains we considered in this study.**
(TXT)

**S4 File. All used domain-combinations present in the antiSMASH-DB dataset after filtering.**
(TXT)

## Acknowledgments

We thank Dr Dick de Ridder and Dr Simon Rogers for useful comments and discussions.

## Author Contributions

**Conceptualization:** Joris J. R. Louwen, Marnix H. Medema, Justin J. J. van der Hooft.

**Data curation:** Joris J. R. Louwen, Satria A. Kautsar.

**Formal analysis:** Joris J. R. Louwen.

**Funding acquisition:** Marnix H. Medema, Justin J. J. van der Hooft.

**Investigation:** Joris J. R. Louwen.

**Methodology:** Joris J. R. Louwen, Sven van der Burg.

**Project administration:** Marnix H. Medema, Justin J. J. van der Hooft.

**Resources:** Marnix H. Medema.

**Software:** Joris J. R. Louwen.

**Supervision:** Satria A. Kautsar, Sven van der Burg, Marnix H. Medema, Justin J. J. van der Hooft.

**Validation:** Joris J. R. Louwen, Sven van der Burg.

**Visualization:** Joris J. R. Louwen.

**Writing – original draft:** Joris J. R. Louwen.

**Writing – review & editing:** Joris J. R. Louwen, Satria A. Kautsar, Sven van der Burg, Marnix H. Medema, Justin J. J. van der Hooft.

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
