## [Decision Letter · Decision Letter 0]

16 Dec 2022

Dear Dr van der Hooft,

Thank you very much for submitting your manuscript "iPRESTO: automated discovery of biosynthetic sub-clusters linked to specific natural product substructures" for consideration at PLOS Computational Biology. As with all papers reviewed by the journal, your manuscript was reviewed by members of the editorial board and by several independent reviewers. The reviewers appreciated the attention to an important topic. Based on the reviews, we are likely to accept this manuscript for publication, providing that you modify the manuscript according to the review recommendations.

Our sincere apologies for the delays in the review process. We had many problems finding available reviewers.

Based the on the two reports received and my own general review, I agree on the usefulness of the tool and I do not think major changes are necessary in the paper.

Sincerely,

Jaime Huerta Cepas

Guest Editor

PLOS Computational Biology

Kiran Patil

Section Editor

PLOS Computational Biology

Dear Dr. van der Hooft,

First of all, our sincere apologies for the delays in the review process. We had many problems finding available reviewers.

Based the on the two reports received and my own general review, I agree on the usefulness of the tool and I do not think major changes are necessary in the paper.

Please address the minor changes suggested by the two reviewers or explain why they cannot be addressed.

best wishes,

-jaime

Reviewer's Responses to Questions

**Comments to the Authors:**

Reviewer #1: The identification of subclusters within predicted BGCs is highly useful for both attempting to link a cryptic BGC to a proposed molecule or for attempts to predict encoded structure in the process of discovery. I believe iPRESTO will make an invaluable contribution to the Antismash framework. The manuscript is well written and I believe it should be accepted for publication. I only have two small comments that I believe will improve the manuscript and the ability of readers to better utilize this study.

- Only 45 subclusters were annotated but I believe the remaining unannotated subclusters should be provided in the excel spreadsheet, perhaps in a different tab. Information on these remaining subclusters may help other researchers with their predictions or may even be annotated by other studies in the future.

- Whilst the authors sometimes say that only inferences are made – I would like to urge the authors to remember that these are in fact only bioinformatic predictions and that direct evidence is lacking, particularly in the earlier parts of the manuscript. Whilst it is stated sometimes, I would like to ask the authors to go through the manuscript and check their language to ensure this aspect is clear.

Reviewer #2: Louwen et al. present iPRESTO, a tool which can explore a database of biosynthetic gene clusters (BGCs) to extract common subcluters. They adapt LDA (latent Dirichlet allocation) for this problem, which improves over the previous state-of-the-art. They show that they can reproduce some known findings and find some suggestive novel assignments for certain metabolites.

Overall, the work is well executed and the value of the approach is well demonstrated. I have only minor comments on how the work is presented.

Minor Comments

#1 The paper is framed as a tool that analyses a collection of BGCs, but to me it seems that most of the value is in the database that the authors explore. On the git repository, it seems that it is possible for a user to map their BCGs into the motifs that the authors have discovered ("Querying existing sub-cluster models"). I estimate that this is much more useful than running the subcluster discovery from scratch and I suggest mentioning it more promeniently in the manuscript. The authors do note that this will be included in antiSMASH in the future, but if the functionality is already available on the command line tool, it could be mentioned in the manuscript.

#2 PRESTO-STAT should be introduced in a way that is understandable even for readers that are not familiar with the prior method on which it is based.

#3 The motivation for why it is a positive thing that many nested sub-clusters are avoided by PRESTO-STAT/PRESTO-TOP should be explained better. IIUC, the problem with previous methods is that it reports too many overlapping/redundant nested sub-clusters. This should be made explicit early.

#4 Line 164: This is unclear to me. If I understand it correctly, the problem is to query a BCG for whether it contains (or approximately contains) any known sub-clusters. Even a fairly naive index would probably work because the number of subclusters that each Pfam domain only participates in is not that large. Perhaps I have misunderstood what was meant.

Line 469: I do not understand the sentence "For a sub-cluster to be considered it needs to consist of more than one gene, for which we set a cut-off of 1.1 on the summed feature probabilities." Which feature probability are being summed and how can their sum exceed 1.0?

Line 183: Figure reference is incomplete, should probably be 2C.

**Have the authors made all data and (if applicable) computational code underlying the findings in their manuscript fully available?**

Reviewer #1: Yes

Reviewer #2: Yes

PLOS authors have the option to publish the peer review history of their article (what does this mean?). If published, this will include your full peer review and any attached files.

Reviewer #1: No

Reviewer #2: No

Figure Files:

Data Requirements:

Reproducibility:

References:

---

## [Decision Letter · Decision Letter 1]

24 Jan 2023

Dear Dr van der Hooft,

We are pleased to inform you that your manuscript 'iPRESTO: automated discovery of biosynthetic sub-clusters linked to specific natural product substructures' has been provisionally accepted for publication in PLOS Computational Biology.

Best regards,

Jaime Huerta Cepas

Guest Editor

PLOS Computational Biology

Kiran Patil

Section Editor

PLOS Computational Biology

Reviewer's Responses to Questions

**Comments to the Authors:**

Reviewer #1: The authors have made fantastic efforts to address the reviewer comments and I think the manuscript is now ready for publication

Reviewer #2: The reviewers have addressed all my concerns.

**Have the authors made all data and (if applicable) computational code underlying the findings in their manuscript fully available?**

Reviewer #1: Yes

Reviewer #2: Yes

PLOS authors have the option to publish the peer review history of their article (what does this mean?). If published, this will include your full peer review and any attached files.

Reviewer #1: No

Reviewer #2: No

---

## [Editor Report · Acceptance letter]

6 Feb 2023

PCOMPBIOL-D-22-01190R1 

iPRESTO: automated discovery of biosynthetic sub-clusters linked to specific natural product substructures

Dear Dr van der Hooft,

I am pleased to inform you that your manuscript has been formally accepted for publication in PLOS Computational Biology. Your manuscript is now with our production department and you will be notified of the publication date in due course.

With kind regards,

Zsofi Zombor
